# Lower Plasma Magnesium, Measured by Nuclear Magnetic Resonance Spectroscopy, is Associated with Increased Risk of Developing Type 2 Diabetes Mellitus in Women: Results from a Dutch Prospective Cohort Study

**DOI:** 10.3390/jcm8020169

**Published:** 2019-02-01

**Authors:** Joëlle C. Schutten, António W. Gomes-Neto, Gerjan Navis, Ron T. Gansevoort, Robin P. F. Dullaart, Jenny E. Kootstra-Ros, Richard M. Danel, Frans Goorman, Rijk O. B. Gans, Martin H. de Borst, Elias J. Jeyarajah, Irina Shalaurova, James D. Otvos, Margery A. Connelly, Stephan J. L. Bakker

**Affiliations:** 1Department of Internal Medicine, Division of Nephrology, University of Groningen, University Medical Center Groningen, Hanzeplein 1, 9700RB Groningen, The Netherlands; a.w.gomes.neto@umcg.nl (A.W.G.-N.); g.j.navis@umcg.nl (G.N.); r.t.gansevoort@umcg.nl (R.T.G.); r.o.b.gans@umcg.nl (R.O.B.G.); m.h.de.borst@umcg.nl (M.H.d.B.); s.j.l.bakker@umcg.nl (S.J.L.B.); 2Department of Endocrinology, University of Groningen, University Medical Center Groningen, Hanzeplein 1, 9700RB Groningen, The Netherlands; r.p.f.dullaart@umcg.nl; 3Department of Laboratory Medicine, University of Groningen, University Medical Center Groningen, Hanzeplein 1, 9700 RB Groningen, The Netherlands; j.e.kootstra@umcg.nl; 4Magnesium Health Institute, 9712 NT Groningen, The Netherlands; danel@soundhealth.onmicrosoft.com; 5Nedmag B.V., 9640 AE Veendam, The Netherlands; f.Goorman@nedmag.nl; 6Laboratory Corporation of America® Holdings (LabCorp), Morrisville, NC 27560, USA; jeyarae@labcorp.com (E.J.J.); shalaui@labcorp.com (I.S.); otvosj@labcorp.com (J.D.O.); connem5@labcorp.com (M.A.C.)

**Keywords:** magnesium, diabetes, nuclear magnetic resonance spectroscopy

## Abstract

Background: Low circulating magnesium (Mg) is associated with an increased risk of developing type 2 diabetes mellitus (T2DM). We aimed to study the performance of a nuclear magnetic resonance (NMR)-based assay that quantifies ionized Mg in EDTA plasma samples and prospectively investigate the association of Mg with the risk of T2DM. Methods: The analytic performance of an NMR-based assay for measuring plasma Mg was evaluated. We studied 5747 subjects free of T2DM at baseline in the Prevention of Renal and Vascular End-stage Disease (PREVEND) study. Results: Passing–Bablok regression analysis, comparing NMR-measured ionized Mg with total Mg measured by the Roche colorimetric assay, produced a correlation of *r* = 0.90, with a slope of 1.08 (95% CI: 1.00–1.13) and an intercept of 0.02 (95% CI: −0.02–0.08). During a median follow-up period of 11.2 (IQR: 7.7–12.0) years, 289 (5.0%) participants developed T2DM. The association of NMR-measured ionized Mg with T2DM risk was modified by sex (P_interaction_ = 0.007). In women, we found an inverse association between Mg and the risk of developing T2DM, independent of adjustment for potential confounders (HR: 1.80; 95% CI: 1.20–2.70). In men, we found no association between Mg and the risk of developing T2DM (HR: 0.90; 95%: 0.67–1.21). Conclusion: Lower NMR-measured plasma ionized Mg was independently associated with a higher risk of developing T2DM in women, but not in men.

## 1. Introduction

The global prevalence of type 2 diabetes mellitus (T2DM) has increased over the past few decades [1], and certain modifiable risk factors, including obesity and insulin resistance, as well as inadequate intake of vitamins and minerals, have received considerable interest [2,3]. Magnesium (Mg) is an essential cofactor for multiple enzymatic pathways involved in energy metabolism and the modulation of insulin-mediated glucose uptake [4] and has been associated with inflammation and endothelial dysfunction [5,6]. Not surprisingly, Mg levels have been linked to several cardiovascular diseases, including ischemic heart disease, stroke, and hypertension, but also to T2DM [7,8,9,10,11,12,13]. 

Nearly 99% of the magnesium in the body is found in the bone, muscle, and soft tissue [14,15]. Only about 0.3–1% is present in serum, with a mean Mg concentration of nearly 0.85 mmol/L. Of this, 70–80% is available in a free ionized form and the rest is bound to proteins, phosphate, citrate, and other compounds. In current clinical laboratories, Mg is measured largely as total Mg with the predominant techniques being (1) photometry, which uses a number of chromogenic substances such as xylidyl blue, and (2) atomic absorption spectroscopy [15]. The determination of ionized Mg has been problematic and ion-selective electrodes for measuring ionized Mg potentiometrically have historically suffered from a lack of selectivity, as well as relatively long response times. In recent years, efforts have been underway to optimize measurement of ionized Mg in plasma and serum due to numerous publications promoting the relevance of ionized Mg in different clinical situations and the potential superiority of ionized Mg over total Mg concentrations [15].

Recently, a clinical nuclear magnetic resonance spectroscopy (NMR) instrument (Vantera^®^ Clinical Analyzer, Morrisville, NC, USA) was developed that addresses the limiting factors of research NMR instruments and allows for the simultaneous quantification of lipoprotein particles, metabolites, and an inflammatory marker in the clinical laboratory [16,17,18,19]. The aim of the current study was to develop and validate an assay for quantifying ionized Mg in plasma using NMR spectra collected for routine lipoprotein quantification on a clinical laboratory instrument. In this way, Mg can be measured in addition to routine lipoprotein quantification without incurring extra costs. With this newly developed NMR-based assay, we further aimed to determine the prospective association of NMR-measured Mg and the risk of developing T2DM in a large Dutch cohort study.

## 2. Materials and Methods

### 2.1. Study Design

For the analyses of the present study, the Prevention of Renal and Vascular Endstage Disease (PREVEND) study was used, which is a prospective Dutch cohort. Details are described elsewhere [20]. In brief, from 1997 to 1998, all inhabitants of Groningen, the Netherlands, aged 28 to 75 years (*n* = 85,421), were sent a short questionnaire on demographic characteristics and renal and cardiovascular morbidity and a vial to collect a first morning void urine sample. Those who were unable or unwilling to participate, pregnant women, and individuals using insulin were not included. Altogether, 40,856 people (48%) responded. Subjects with a urinary albumin concentration of ≥10 mg/L (*n* = 7768) were invited to participate, of whom 6000 subjects were enrolled. In addition, a randomly selected group with a urinary albumin concentration of <10 mg/L (*n* = 3394) was invited to participate in the cohort and 2592 subjects of the initially 3394 invited subjects were enrolled. 8592 subjects participated in the PREVEND cohort and completed an extensive examination in 1997 and 1998 (baseline). Participants were invited to the outpatient clinic of the University Medical Center Groningen for measurements approximately every 3 years. 

The second screening took place from 2001 through 2003 (*n* = 6894), which was the starting point for the present evaluation. For the present study, we excluded subjects with diabetes at baseline or unknown diabetes status or with no follow-up data available for diabetes (*n* = 545) and subjects with missing Mg data (*n* = 602), leaving 5747 participants for the analyses (Figure 1). The PREVEND study was approved by the Medical Ethics Committee of the University Medical Center Groningen. Written informed consent was obtained from all participants and was performed according to the principles outlined in the Declaration of Helsinki. All participants provided written informed consent. 

### 2.2. Laboratory Analysis

Venous blood was obtained at each screening round after an overnight fast. EDTA plasma and lithium heparin plasma samples were prepared by centrifugation at 4 °C and stored at −80 °C until thawed for testing. EDTA plasma samples from the second screening were sent frozen to LipoScience, (now LabCorp, Morrisville, NC, USA) for testing on the Vantera Clinical Analyzer (Morrisville, NC, USA) and lithium heparin plasma was tested on the Roche Modular system.

### 2.3. NMR-Based Ionized Mg Assay

As is customary for collecting NMR spectra for the NMR LipoProfile test, EDTA plasma samples were diluted 1:1 with phosphate buffer (pH 7.4) containing 5 mmol/L EDTA. The extra EDTA in the buffer ensured complete chelation of free ionized Mg present in the plasma specimens, as well as any circulating Mg that may not be tightly bound to proteins, citrates, or phosphates. Proton NMR spectra were collected on 400 MHz Vantera Clinical Analyzers at 47 °C as described previously [16,21]. The NMR acquisition time was 48 s, with a total sample to sample turnaround time of 90 s. The proton NMR spectra were deconvoluted using proprietary software as follows: The singlet peak emanating from four equivalent protons of the ethylene moiety in the Mg-EDTA complex (-N-CH2-CH2-N-) appearing at 2.66 ppm in the NMR spectrum was used for quantitation. As the Mg-EDTA NMR signal overlaps with a signal from circulating proteins, the deconvolution method included the protein signal encompassing approximately 50 Hz, and a 16 Hz wide region of the Mg-EDTA peak was integrated. The relation between Mg-EDTA signal area and Mg concentrations were established by standard addition experiments on dialyzed serum, and the conversion factor thus obtained was applied to transform Mg-EDTA signal areas to concentrations expressed in mmol/L. The Mg concentrations were standardized against a 25.0 mM solution of ACS Reagent Grade MgCl_2_.6H_2_O (MilliporeSigma, Burlington, MA, USA). Defined amounts of the standard MgCl_2_ solution were spiked into dialyzed serum devoid of ionized Mg. The accuracy was ascertained through recovery experiments done from 0 to 4.0 mM Mg concentrations. Similar to other NMR assays on Vantera, the commercial assay would also involve running 2 levels of serum controls, performing daily checks on the accuracy, and guarding against drift with time. We tested for imprecision in the NMR-measured ionized Mg assay as per CLSI guidelines. Pooled samples with two varying concentrations of Mg (low and high) were tested to determine within-lab (inter-assay) precision.

### 2.4. Roche Modular Total Mg Assay and Assay Comparison

The Roche Modular assay is a colorimetric end point assay that measures total Mg in a serum, heparin plasma or urine sample. The method is based on the reaction of Mg with xylidyl blue in an alkaline solution containing ethylene glycol-bis(β-aminoethyl ether)-N,N,N′,N′-tetraacetic acid (EGTA), which has a lower affinity for Mg, in order to mask the calcium in the sample. In the alkaline solution, Mg forms a purple complex with the xylidyl blue diazonium salt and the concentration of Mg is determined photometrically via the decrease in the xylidyl blue absorbance (505/600 nm). In order to understand the differences between the two assays, we compared values from the NMR-based ionized Mg assay with total Mg measured on a Roche Modular Analyzer (Roche Diagnostics, Mannheim, Germany) in 799 samples of appropriate specimen types from the second screening of the PREVEND cohort. The Roche Mg assay has an inter-assay coefficient variation of 1.3%. 

### 2.5. Assessment of Covariates

Body mass index (BMI) was calculated as weight (kg) divided by height squared (m^2^). Smoking status was defined as self-reported never smoker, former smoker, or current smoker [22]. Blood pressure was measured with an automatic Dinamap XL Model 9300 series device (Johnson-Johnson Medical, Tampa, FL, USA). Hypertension was defined as a systolic blood pressure (SBP) > 140 mmHg or a diastolic blood pressure (DPB) > 90 mmHg, and/or the use of anti-hypertensive drugs. Information on medication use was combined with information on drug use from the IADB.nl database, containing pharmacy-dispensing data from community pharmacies in the Netherlands [23]. Estimated glomerular filtration rate (eGFR) based on serum creatinine and serum cystatin C was calculated from the Chronic Kidney Disease Epidemiology Collaboration equation [24]. Urinary albumin, sodium, urea, and creatinine excretion and circulating albumin, sodium, potassium, calcium, and creatinine, total cholesterol, high-density lipoprotein cholesterol, triglycerides, high sensitivity C-reactive protein (hsCRP) and glucose were determined as previously described [25,26,27,28]. 

### 2.6. Assessment of T2DM Risk

Incident T2DM was ascertained if one or more of the following criteria were met: (1) Fasting plasma glucose > 7.0 mmol/L; (2) random sample plasma glucose > 11.1 mmol/L; (3) self-reporting of a physician diagnosis; (4) initiation of glucose-lowering medication use retrieved from a central pharmacy registry [29]. Incident T2DM was defined as T2DM that occurred after the second screening.

### 2.7. Statistical Analysis 

Analytic validation data was calculated using Analyze-it (Analyze-it Software, Ltd. Leeds, UK). Passing–Bablok regression analysis was used to test agreement between NMR-measured Mg in EDTA plasma and heparin plasma Mg measured on the Roche Modular Analyzer. Bland-Altman plots were used to visualize bias. Baseline characteristics are reported in terms of means (SD) when normally distributed or medians (interquartile range) in the case of non-normally distributed data. Categorical data are presented as frequencies (percentages). We prospectively examined the association between chelated Mg and the risk of developing T2DM using Cox proportional hazards regression models. We used chelated Mg as a continuous variable in these models and additionally, we examined the association in tertiles of chelated Mg. Person-time of follow-up was calculated for each participant from the first visit (baseline) until the last visit, the incidence of T2DM, death, or relocation to an unknown destination, whichever came first. Multivariable Cox models were adjusted for age, sex, BMI, smoking (2 categories), alcohol intake (2 categories), triglyceride to high-density lipoprotein cholesterol ratio, hypertensive treatment, parental history of T2DM, plasma levels of albumin, potassium and calcium and urinary albumin excretion, and fasting glucose levels, CRP, and eGFR. In addition, we performed sensitivity analyses in which we replaced adjustment for antihypertensive treatment in Cox regression analyses with an adjustment for the presence of cardiovascular disease and in which we replaced the adjustment for eGFR in Cox regression analyses with an adjustment for the presence of chronic kidney disease. Hazard ratios (HRs) are reported with 95% confidence intervals (CIs). Restricted cubic splines with three knots were performed to show the association between ionized Mg and risk of T2DM using Cox regression analyses. We evaluated the potential effect modification in the analyses of plasma ionized Mg and risk of T2DM by fitting models containing both main effects and their cross-product terms. The interaction terms were considered statistically significant at *p* < 0.10. Missing data (present in 0.0–13.4%) in covariables were handled by multiple imputations [30]. Results are reported for imputed data, except for the baseline characteristics and the analytic validation data. We considered a two-sided *p* value < 0.05 as statistically significant. Data were analyzed using SPSS Statistics version 23.0 (SPSS Inc, Chicago, IL, USA). 

## 3. Results

### 3.1. Analytical Performance of the NMR-Measured Ionized Mg Assay

The coefficient of variation for the NMR ionized Mg assay ranged from 4.6% to 7.1% for within-lab imprecision (Table 1). We compared NMR-measured Mg in EDTA plasma specimens with lithium heparin plasma total Mg measured by a Roche Modular colorimetric assay in 799 samples from the PREVEND study. We found a strong linear relationship between NMR-measured ionized Mg and colorimetrically measured total Mg (*r* = 0.90). Bland–Altman analysis showed a systematic bias of 0.07 mmol/L with chemically measured total Mg concentrations being slightly higher than the ionized Mg quantified by NMR (Figure 2). Passing–Bablok regression analysis revealed an intercept of 0.02 (95% CI: −0.02–0.08) and a slope of 1.08 (95% CI: 1.00–1.13) (Figure 3). 

Two pools of EDTA plasma with low and high Mg concentrations were tested twice a day in duplicate for 20 days on one instrument.

### 3.2. Association of Ionized Mg with the Risk of Developing T2DM 

The baseline characteristics of the 5747 participants are shown in Table 2. Mean age was 53.0 ± 11.9 years, mean NMR-measured ionized Mg was 0.75 ± 0.05 mmol/L, and 50.4% of the participants were female. Higher ionized Mg was associated with a slightly lower BMI, lower fasting glucose levels and higher HDL-cholesterol levels. In addition, subjects in the highest tertile of ionized Mg were more likely to be non-smokers. 

During a median follow-up period of 11.2 (IQR: 7.7–12.0) years, 289 (5.0%) participants developed T2DM. We found an association between the levels of NMR-measured ionized Mg and the risk of developing T2DM in the total population (HR: 1.50; 95% CI: 1.19–1.89). However, after multivariable adjustment, the association lost significance (HR: 1.16; 95% CI: 0.91–1.47). The association of NMR-measured Mg with the risk of T2DM was modified by sex (P_interaction_ = 0.007). In men, the association between NMR-measured Mg and the risk of developing T2DM was non-significant in the crude model (HR: 1.25; 95% CI: 0.94–1.67) and in a fully adjusted multivariable model (HR: 0.90; 95%: 0.67–1.21) (Table 3). In women, on the other hand, we found an association between NMR-measured Mg and the risk of developing T2DM in the crude model (HR: 2.02, 95% CI: 1.37–2.99). After adjustment for lifestyle factors, including BMI, alcohol consumption, smoking status, triglyceride to HDL cholesterol ratio, use of antihypertensive drugs, and parental history of T2DM (Model 1), the hazard ratio was slightly attenuated, but not substantially different (HR: 1.66; 95% CI: 1.11–2.47) (Table 3). When we further adjusted for the variables in Model 3, including fasting glucose, CRP, and eGFR, the association remained similar (HR: 1.80; 95% CI: 1.20–2.70). Furthermore, the hazard ratio for women in the lowest tertile of NMR-measured Mg was 1.65 (95% CI: 1.02–2.66) in the crude analysis and 1.72 (95% CI: 1.03–2.86) after multivariable adjustment. A restricted spline curve confirmed the log-linear inverse association of NMR-measured Mg and the risk of developing T2DM in women (Figure 4). Sensitivity analyses yielded similar results as in our main analyses (Table 4). 

## 4. Discussion 

This is the first study showing that accurate concentrations of NMR-measured ionized Mg can be obtained from NMR spectra collected on EDTA plasma samples tested for routine lipoprotein quantification on a clinical NMR analyzer. The ability to simultaneously interrogate disease associations for lipoprotein particles, small molecule metabolites, and the GlycA inflammatory markers in large epidemiological studies using analytically and clinically validated assays is one of the strengths of the Vantera NMR Clinical Analyzer platform. As a standalone test, the cost for NMR-assay for Mg would be comparable to a lipid panel derived from the NMR assay. If ordered together with NMR LipoProfile test or NMR Extended Lipid Panel test, it would only add a small incremental cost since no additional NMR time would be needed. Unlike the chemical assays for Mg, the non-invasive NMR assay does not require reagents, and one can get results for all of the NMR based tests from one NMR spectra obtained from a single draw of a specimen, adding to the overall cost saving. It is therefore of particular interest for research when samples are precious and available sample volume relatively low, as for instance often is the case in cohort studies and intervention studies using the material of underlying biobanks. The assay is currently available through the NMR Global Research Services group at LabCorp for research use only. Investigators who have used the NMR LipoProfile test over the years for various cardiovascular, diabetes, nutrition and diet, and inflammation-related research studies and clinical trials have the opportunity of retrospectively analyzing the stored NMR data for plasma Mg levels. Though the data presented in this paper pertains to EDTA plasma specimens, the NMR-assay for Mg can easily be adapted to work on serum specimens by sufficiently modifying the EDTA concentration in the NMR diluent.

The systematic bias of 0.07 mmol/L, as shown in Bland–Altman plots, is likely due to the differences in the amount of Mg that is quantified in each of the assays; the NMR assay quantifying largely free ionized Mg and the Roche Modular colorimetric assay measuring total Mg levels. Similar results were reported in a paper by Koch et al., who compared total Mg with free ionized Mg, suggesting that ionized Mg measured by NMR and ionized Mg quantified by ion specific electrode measurement may be similar in the fraction of circulating Mg that they are able to quantify [31]. Moreover, the Roche method can only use serum or heparinized plasma as the preferred specimens but not EDTA plasma. In contrast, the NMR method requires the use of EDTA plasma specimens only. The difference in the types of specimen used for the two assays in this study (EDTA vs. lithium heparin) may also contribute to the small difference in the amount of Mg that was observed.

Our prospective findings showed that NMR-measured ionized Mg was associated with an increased risk of T2DM in women even after adjusting for traditional T2DM risk factors and CRP. However, we found no association between NMR-measured Mg and the risk of developing T2DM in men. No other studies have reported such an interaction with sex in the association of circulating Mg and the risk of T2DM. This is, therefore, the first study demonstrating an inverse association of Mg on T2DM risk that is only present in women. The interaction between circulating Mg levels and sex may warrant confirmation in further studies. It might be that female sex hormones play a role in this observed interaction [32]. It has indeed been established that estrogen significantly affects renal magnesium handling [33,34], likely explaining higher circulating Mg levels in pre-menopausal women than in post-menopausal women and the cycling of Mg levels in pre-menopausal women [35,36]. Low circulating Mg levels and T2DM are known to be related, mainly through insulin resistance rather than through insulin secretion, but the cause and effect relationships remain to be established [10]. Consistent with this, it has been observed that low circulating Mg levels are associated with diabetes, insulin resistance, and obesity in women, but not in men [37]. It has also been shown that there is a sex difference in the relationship of urinary magnesium excretion to glycaemic control in patients with T1DM [38]. Albeit not in the field of diabetes, it has repeatedly been suggested that there is a sex difference in the prospective association of Mg intake and low circulating Mg levels with cardiovascular mortality, with associations predominantly present in or limited to women [39,40,41]. 

So far, only three studies prospectively reported associations between serum or plasma Mg and the risk of developing T2DM [42,43,44]. Everett et al. investigated the relationship between circulating Mg and the risk of developing T2DM in a cohort of 9784 US participants [43]. They showed that low serum Mg was associated with an increased risk of T2DM in the total population. Recently, Kieboom et al. confirmed this finding and showed that the association was partly mediated through insulin resistance [44]. In the present study, the association between NMR-measured Mg and T2DM remained significant after adjusting for fasting glucose levels, which is also in line with the findings of Kao et al. [42]. Indeed, the effect of Mg on insulin actions could be one possible mechanism by which Mg affects T2DM risk. Mg is essential for autophosphorylation of the β-subunits of the insulin receptor; in vitro studies have shown that Mg enhances tyrosine kinase activity by increasing the receptor’s affinity for ATP. Thus, intracellular Mg deficiency may result in decreased tyrosine kinase activity and consequently in insulin resistance. Furthermore, clinical trials have shown that Mg supplementation is effective in reducing fasting plasma glucose in diabetic patients and HOMA-IR in individuals at risk of T2DM [45]. 

Another possible explanation by which low Mg levels affect the risk of T2DM might be through chronic inflammation. Markers for systemic inflammation, including CRP and GlycA, have been found to be independent predictors for the risk of developing T2DM [28,46]. Song and others found that dietary Mg intake was inversely associated with CRP and E-selectin levels in women and recently, it has been shown that low circulating Mg levels were associated with higher CRP levels [47]. Furthermore, a meta-analysis of clinical trials showed that Mg supplementation reduces CRP levels, but this finding was only significant in subjects with higher baseline CRP levels [48]. In our study, adjustment for CRP did not influence the association between plasma Mg and T2DM risk.

Measurements of circulating Mg are currently used by physicians to identify patients with hypomagnesemia and hypermagnesemia. Hypermagnesemia or high magnesium may be an indication of renal impairment or failure or in patients with Addison disease. Marked increases in circulating Mg may also be found in patients taking Mg salts, such as those found in antacids, or in pregnant women with preeclampsia who are taking Mg sulfate as an anticonvulsant. Hypomagnesemia can occur in subjects who are on long-term hyperalimentation, intravenous therapy, or suffering from alcoholism and other types of malnutrition or malabsorption. Mg deficiency has been shown to be associated with cardiac arrhythmias. This study suggests that low plasma Mg may also be useful for assessing the risk of T2DM in women.

The present study has strengths as well as limitations. The main strengths include the prospective study design and the large sample size. Second, our findings suggest that NMR-measured ionized Mg showed good agreement with Mg measured on the Roche Modular Analyzer, thereby, introducing a novel method to measure circulating Mg that might have useful clinical applications. Some limitations warrant consideration. First, as with any observational study, no cause-effect relationships can be drawn and residual confounding still may exist. Second, because Mg is predominantly present in green leafy vegetables, nuts, whole grains, and legumes, the effects of other dietary components could have been responsible for the association between Mg and T2DM. Unfortunately, we were not able to adjust for such potential confounders, because no data on dietary intake were available. Third, we had to report separate HRs for men and women, because we found a significant difference in interaction by sex in the analysis. Therefore, the number of cases could have become insufficient to detect an association in men. Fourth, NMR-measured Mg was only assessed at the second screening and, therefore, we could not take into account possible changes in Mg concentrations over time. However, one study reported a strong correlation between two Mg concentrations that were measured 1 year apart [49]. Finally, because >95% of the subjects within the PREVEND study are white and of Dutch origin, the results may not be generalizable to different ethnic populations. 

## 5. Conclusions

In summary, ionized Mg quantified in the Vantera Clinical Analyzer showed good agreement with total Mg measured on the Roche Modular Analyzer. In prospective analyses, we found that lower NMR-measured ionized Mg is associated with a higher risk of developing T2DM in women, but not in men. This association was independent of several traditional T2DM risk factors. These findings warrant future long-term clinical trials to study thoroughly the potential effects of Mg supplementation on prevention of T2DM and T2DM control. In addition, future studies should focus on the interaction between Mg and sex in the association with T2DM. 

## Figures and Tables

**Figure 1 jcm-08-00169-f001:**
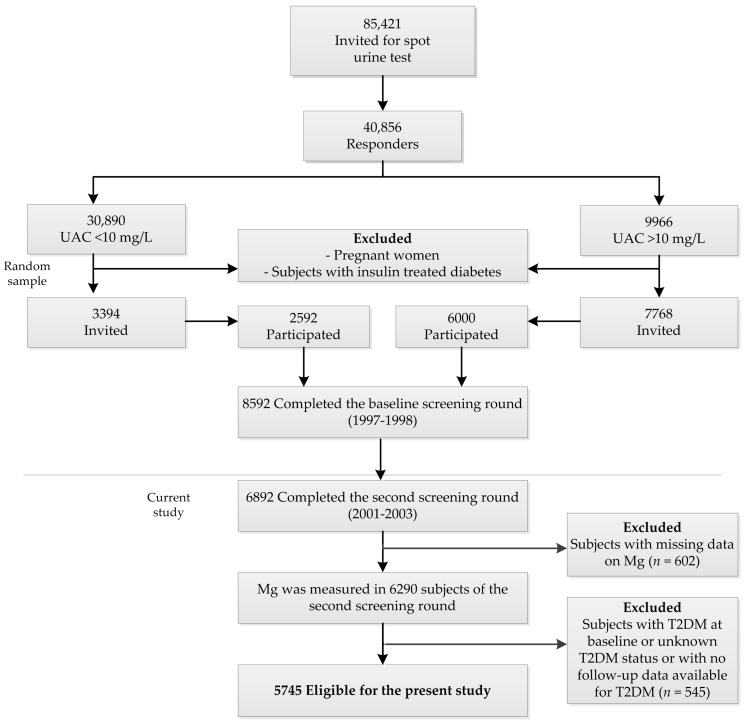
Flowchart of the Prevention of Renal and Vascular Endstage Disease (PREVEND) study participants included or excluded for the purposes of this study.

**Figure 2 jcm-08-00169-f002:**
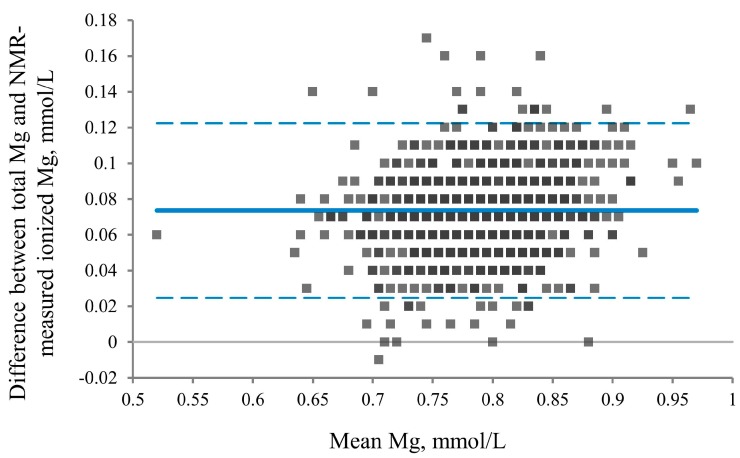
Bias plots for total Mg from the Roche Modular and nuclear magnetic resonance (NMR)-measured ionized Mg.

**Figure 3 jcm-08-00169-f003:**
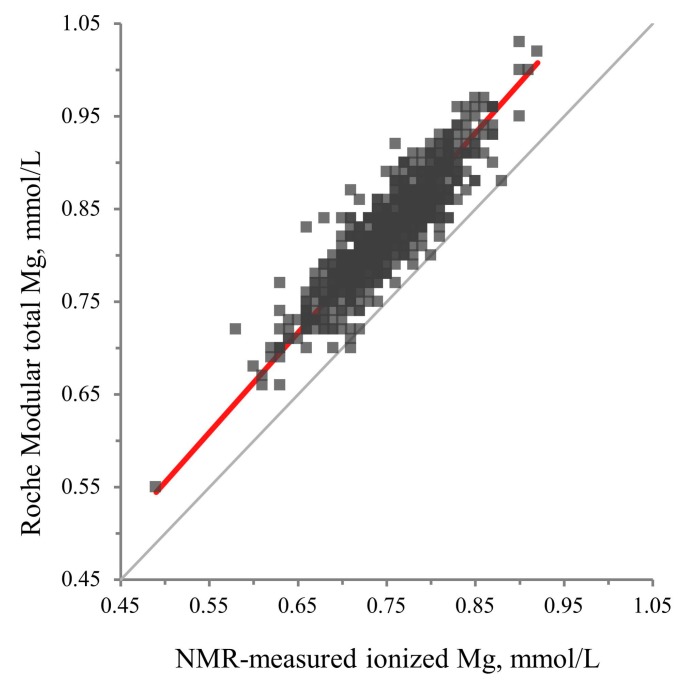
Passing–Bablok regression analysis.

**Figure 4 jcm-08-00169-f004:**
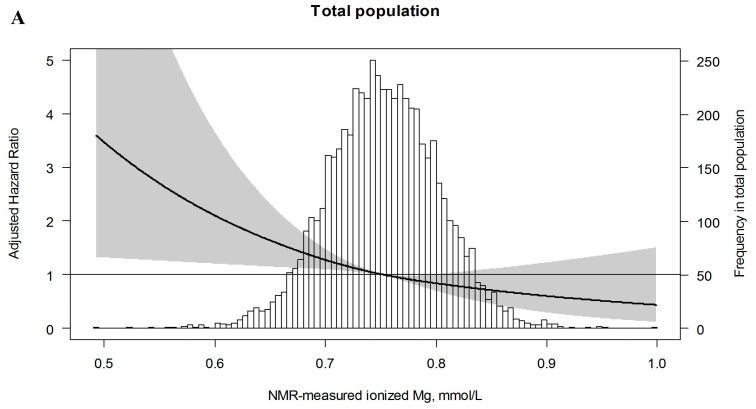
Association between NMR-measured Mg and the risk of developing T2DM. Data were fit by a restricted cubic spline with 3 knots and were adjusted for age (**A–C**) and sex (**A**).

**Table 1 jcm-08-00169-t001:** Within-lab imprecision of ionized Mg measured on the Vantera Clinical Analyzer.

Within-Lab	NMR-Measured Mg (mmol/L)	
	Low	High
Mean	0.489	0.892
SD	0.035	0.041
CV	7.1%	4.6%

**Table 2 jcm-08-00169-t002:** Baseline characteristics of the PREVEND study population.

	Tertiles of NMR-Measured Mg, mmol/L			*P* Trend
	1 (<0.73)	2 (0.73–0.77)	3 (>0.77)	
Participants, *n*	1919	1900	1928	-
Age, years	52.3 ± 11.8	52.2 ± 11.7	54.4 ± 12.0	<0.001
Female, %	48.7	52.1	50.5	0.27
Race, whites, %	96.2	95.9	96.0	0.65
Body mass index, kg/m^2^	26.9 ± 4.3	26.5 ± 4.3	26.1 ± 3.9	<0.001
Smoking status, %				<0.001
Never	25.9	29.3	30.9	
Former	42.2	43.1	43.2	
Current	31.9	27.6	25.9	
Alcohol consumption, %				0.82
None	24.1	23.8	23.5	
1–4 drinks per month	16.1	17.7	17.4	
2–7 drinks per week	31.9	32.6	31.4	
1–3 drinks per day	21.7	21.9	22.3	
4 or more drinks per day	5.3	3.3	4.2	
Education, %				0.17
Low	43.3	41.6	42.4	
Middle	26.4	27.2	24.3	
High	30.4	31.2	33.3	
Glucose, mmol/L	4.89 ± 0.66	4.82 ± 0.65	4.81 ± 0.63	<0.001
Parental history of T2DM, %	14.0	14.6	14.6	0.61
Blood pressure, mm Hg				
Systolic	125.7 ± 18.2	124.0 ± 17.2	126.2 ± 19.2	0.43
Diastolic	73.4 ± 9.1	72.7 ± 8.9	73.3 ± 9.1	0.87
Hypertension, yes, %	33.1	28.1	32.9	0.90
Use of antihypertensive drugs				
ACEi, %	6.2	5.5	5.2	0.41
ARB, %	2.1	1.5	2.0	0.36
Diuretics, %	6.0	5.3	5.4	0.59
Beta blockers, %	11.1	8.8	8.5	0.01
Total cholesterol, mmol/L	5.40 ± 1.04	5.40 ± 1.03	5.50 ± 1.03	0.004
HDL-cholesterol, mmol/L	1.23 ± 0.29	1.26 ± 0.30	1.30 ± 0.25	<0.001
Triglycerides, mmol/L	1.13 (0.83–1.66)	1.09 (0.78–1.54)	1.08 (0.79–1.55)	0.001
Triglyeride:HDL-cholesterol ratio	2.19 (1.40–3.46)	2.00 (1.33–3.18)	1.93 (1.24–3.17)	<0.001
Use of lipid lowering drugs, yes, %	6.5	7.3	7.6	0.03
CRP, mg/L	1.41 (0.64–3.14)	1.26 (0.58–2.77)	1.26 (0.60–2.86)	0.02
Creatinine, μmol/L	70.0 (61.0–79.0)	71.0 (62.0–80.0)	72.0 (64.0–81.0)	<0.001
Cystatine C, mg/L	0.87 (0.78–0.98)	0.86 (0.78–0.96)	0.89 (0.80–0.99)	<0.001
Estimated GFR, mL/min/1.73^2^	93.6 ± 17.0	94.1 ± 16.2	90.3 ± 17.0	<0.001
Plasma levels of				
Albumin, g/L	43.3 ± 2.8	43.8 ± 2.6	44.2 ± 2.9	<0.001
Sodium, mmol/L	140.5 ± 2.0	140.7 ± 2.0	140.9 ± 2.1	<0.001
Potassium, mmol/L	4.20 ± 0.28	4.22 ± 0.26	4.25 ± 0.29	<0.001
Calcium, mmol/L	2.30 ± 0.12	2.30 ± 0.10	2.30 ± 0.11	0.06
Urinary excretions of				
Albumin, mg/24-h	9.0 (6.1–17.3)	8.2 (5.9–14.0)	8.3 (6.0–13.5)	<0.001
Sodium, mmol/24-h	148.5 ± 56.3	144.3 ± 55.3	139.9 ± 53.5	<0.001
Urea, mmol/24-h	365.6 ± 112.8	367.8 ± 115.0	359.2 ± 110.8	0.07
Creatinine, mmol/24-h	12.7 ± 3.4	12.4 ± 3.3	12.2 ± 3.4	<0.001

Values are presented as means with SDs, medians with interquartile ranges, or percentages. Values are shown for non-imputed data. T2DM, type 2 diabetes mellitus; GFR indicates glomerular filtration rate; HDL, high-density lipoprotein; ACEi, angiotensin converting enzyme inhibitor; ARB, angiotensin receptor blockers; CRP, high sensitive C-reactive protein.

**Table 3 jcm-08-00169-t003:** The association of NMR-measured ionized Mg with the risk of developing type 2 diabetes mellitus (T2DM) in the PREVEND study.

	Continuous, per 0.1 mmol/L Decrease	Tertiles of NMR-Measured Mg, mmol/L		
		1	2	3
Total (*n* = 5747)				
Events, n (%)	289 (5.0)	108 (5.6)	99 (5.2)	82 (4.3)
Crude analysis	1.50 (1.19–1.89)	1.36 (1.02–1.82)	1.23 (1.02–1.82)	1.00 (reference)
Age and sex adjusted	1.54 (1.23–1.92)	1.46 (1.09–1.94)	1.36 (1.01–1.82)	1.00 (reference)
Model 1	1.27 (1.00–1.61)	1.17 (0.87–1.56)	1.19 (0.89–1.60)	1.00 (reference)
Model 2	1.32 (1.04–1.67)	1.20 (0.89–1.61)	1.20 (0.89–1.62)	1.00 (reference)
Model 3	1.16 (0.91–1.47)	1.09 (0.81–1.47)	1.23 (0.91–1.65)	1.00 (reference)
Men (*n* = 2848)				
Events, n (%)	186 (6.5)	66 (6.7)	66 (7.3)	54 (5.7)
Crude analysis	1.25 (0.94–1.67)	1.21 (0.84–1.73)	1.31 (0.91–1.87)	1.00 (reference)
Age adjusted	1.27 (0.96–1.69)	1.26 (0.88–1.80)	1.41 (0.98–2.02)	1.00 (reference)
Model 1	1.04 (0.78–1.40)	1.01 (0.70–1.45)	1.31 (0.92–1.89)	1.00 (reference)
Model 2	1.04 (0.78–1.40)	1.01 (0.70–1.45)	1.31 (0.91–1.89)	1.00 (reference)
Model 3	0.90 (0.67–1.21)	0.85 (0.58–1.24)	1.26 (0.88–1.81)	1.00 (reference)
Women (*n* = 2899)				
Events, n (%)	103 (3.6)	42 (4.5)	33 (3.3)	28 (2.9)
Crude analysis	2.02 (1.37–2.99)	1.65 (1.02–2.66)	1.15 (0.70–1.91)	1.00 (reference)
Age adjusted	2.33 (1.58–3.42)	1.99 (1.23–3.22)	1.28 (0.78–2.13)	1.00 (reference)
Model 1	1.66 (1.11–2.47)	1.45 (0.88–2.39)	1.13 (0.68–1.89)	1.00 (reference)
Model 2	1.88 (1.26–2.79)	1.67 (1.01–2.77)	1.23 (0.73–2.08)	1.00 (reference)
Model 3	1.80 (1.20–2.70)	1.72 (1.03–2.86)	1.30 (0.76–2.20)	1.00 (reference)

Hazard ratios and 95% confidence intervals were derived from Cox proportional hazards regression models. T2DM, type 2 diabetes mellitus; GFR indicates glomerular filtration rate; HDL, high-density lipoprotein. Model 1: Adjusted for age, sex, body mass index, alcohol consumption, smoking status, triglyceride:HDL cholesterol ratio, antihypertensive treatment, and parental history of T2DM. Model 2: Model 1 and additionally adjusted for plasma levels of albumin, potassium, and calcium and urinary albumin excretion. Model 3: Model 2 and additionally adjusted for CRP, fasting glucose, and eGFR.

**Table 4 jcm-08-00169-t004:** The association of NMR-measured ionized Mg with the risk of developing T2DM in the PREVEND study (sensitivity analyses).

	Continuous, per 0.1 mmol/L decrease	Tertiles of NMR-Measured Mg, mmol/L		
		1	2	3
Total (*n* = 5747)				
Events, n (%)	289 (5.0)	108 (5.6)	99 (5.2)	82 (4.3)
Crude analysis	1.50 (1.19–1.89)	1.36 (1.02–1.82)	1.23 (1.02–1.82)	1.00 (reference)
Age and sex adjusted	1.54 (1.23–1.92)	1.46 (1.09–1.94)	1.36 (1.01–1.82)	1.00 (reference)
Model 1	1.32 (1.04–1.67)	1.21 (0.90–1.62)	1.18 (0.90–1.62)	1.00 (reference)
Model 2	1.37 (1.08–1.74)	1.26 (0.93–1.69)	1.20 (0.89–1.62)	1.00 (reference)
Model 3	1.19 (0.94–1.51)	1.12 (0.83–1.51)	1.21 (0.90–1.63)	1.00 (reference)
Men (*n* = 2848)				
Events, n (%)	186 (6.5)	66 (6.7)	66 (7.3)	54 (5.7)
Crude analysis	1.25 (0.94–1.67)	1.21 (0.84–1.73)	1.31 (0.91–1.87)	1.00 (reference)
Age adjusted	1.27 (0.96–1.69)	1.26 (0.88–1.80)	1.41 (0.98–2.02)	1.00 (reference)
Model 1	1.05 (0.78–1.40)	1.01 (0.70–1.45)	1.32 (0.92–1.90)	1.00 (reference)
Model 2	1.04 (0.78–1.41)	1.01 (0.70–1.45)	1.32 (0.92–1.89)	1.00 (reference)
Model 3	0.89 (0.67–1.20)	0.84 (0.58–1.22)	1.26 (0.87–1.81)	1.00 (reference)
Women (*n* = 2899)				
Events, n (%)	103 (3.6)	42 (4.5)	33 (3.3)	28 (2.9)
Crude analysis	2.02 (1.37–2.99)	1.65 (1.02–2.66)	1.15 (0.70–1.91)	1.00 (reference)
Age adjusted	2.33 (1.58–3.42)	1.99 (1.23–3.22)	1.28 (0.78–2.13)	1.00 (reference)
Model 1	1.70 (1.14–2.52)	1.52 (0.92–2.49)	1.12 (0.67–1.87)	1.00 (reference)
Model 2	1.89 (1.27–2.81)	1.71 (1.04–2.83)	1.18 (0.70–1.99)	1.00 (reference)
Model 3	1.81 (1.22–2.69)	1.75 (1.06–2.89)	1.29 (1.06–2.17)	1.00 (reference)

Hazard ratios and 95% confidence intervals were derived from Cox proportional hazards regression models. T2DM, type 2 diabetes mellitus; GFR indicates glomerular filtration rate; HDL, high-density lipoprotein. Model 1: Adjusted for age, sex, body mass index, alcohol consumption, smoking status, triglyceride:HDL cholesterol ratio, presence of cardiovascular disease, and parental history of T2DM. Model 2: Model 1 and additionally adjusted for plasma levels of albumin, potassium, and calcium and urinary albumin excretion. Model 3: Model 2 and additionally adjusted for CRP, fasting glucose, and presence of chronic kidney disease.

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
