# Peer review of "Lower Plasma Magnesium, Measured by Nuclear Magnetic Resonance Spectroscopy, is Associated with Increased Risk of Developing Type 2 Diabetes Mellitus in Women: Results from a Dutch Prospective Cohort Study"

_jcm, 2019, doi:10.3390/jcm8020169_

Reviewer 1 Report

Provide a consort diagram to demonstrate who from the original cohort was included in the final analysis.

Describe inclusion/exclusion criteria.

Table 2-all characteristics should be defined-alcohol intake, education level, antihypertensive treatment

Table 2-Clarify "Hypertension, %" (with or without) and "Lipid lowering drugs, %" (% taking or not taking?)

How were co-variates chosen for the analyses in Table 3? Why were presence of chronic diseases not included (CVD, CKD). It seems instead of presence of these diseases, treatment of them (i.e. antihypertensive treatment, lipid lowering drugs) and lab values (i.e. estimated GFR, creatinine, lipids) were used. Please explain why this method was chosen and possible implications to the findings.

Discussion-I would like to see more discussion about the pros ad cons of the use of this technique in the future. Should it be used and why or why not? What is the availability of this test? cost? What could it mean to the future of magnesium research, which is lacking an accurate, non-invasive test for magnesium status.

Inclusion/exclusion criteria may identify additional limitations based on the study population included (i.e. Dutch population only)

Conclusion-What are the authors recommendations in terms of use of this technique for identifying magnesium status in the future?

Author Response

Question 1
Provide a consort diagram to demonstrate who from the original cohort was included in the final analysis.

Response: Thank you for this suggestion. Accordingly, we provided a consort diagram within the manuscript (Figure 1).

Question 2
Describe inclusion/exclusion criteria.

Response: We agree that description of the inclusion and exclusion criteria could have been morecomplete. Accordingly, we added a more detailed description of the inclusion/exclusion criteria by adding the following sentence: “Those who were unable or unwilling to participate, pregnant women, and individuals using insulin were not allowed to participate” to the Methods section (Line 85-87) of the revised version of the manuscript. In addition, we provided a consort diagram of the study population (Figure 1).
Question 3
Table 2-all characteristics should be defined-alcohol intake, education level, antihypertensive treatment.

Response: To accommodate the comment of the reviewer, we added the categories of 4 characteristics in Table 2, including alcohol intake (5 categories; none, 1-4 drinks per month, 2-7 drinks per week, 1-3 drinks per day, 4 or more drinks per day), smoking behavior (3 categories; never, former and current), educational level (3 categories; low, middle and high) and antihypertensive medication (4 categories; ACEi, ABR, diuretics and beta blockers).  

Question 4
Table 2-Clarify "Hypertension, %" (with or without) and "Lipid lowering drugs, %" (% taking or not taking?) .

Response: To accommodate the comment of the reviewer, we clarified the percentage of subjects with hypertension and the percentage of subjects with lipid lowering drugs by adding “yes” behind hypertension and lipid lowering drug in Table 2.

Question 5
How were co-variates chosen for the analyses in Table 3? Why were presence of chronic diseases not included (CVD, CKD). It seems instead of presence of these diseases, treatment of them (i.e. antihypertensive treatment, lipid lowering drugs) and lab values (i.e. estimated GFR, creatinine, lipids) were used. Please explain why this method was chosen and possible implications to the findings.

Response: Covariates for the Cox Proportional-Hazards models were chosen based on literature and on univariate associations with NMR magnesium that we found in our Baseline table (Table 2). To accommodate the comment of the reviewer, we have performed sensitivity analyses in which we replaced adjustment for antihypertensive treatment in Cox regression analyses by adjustment for CVD and in which we replaced adjustment for eGFR in Cox regression analyses by adjustment for CKD (Table 4). These adjustments for the presence of CVD and CKD did not materially change our findings. To further accommodate the comment of the reviewer, we have added mentioning of performance of these sensitivity analyses to the methods section of the revised version of the manuscript and added mentioning of the results of these analyses to the results section of the revised version of the manuscript (Lines 184-187).

Question 6
Discussion-I would like to see more discussion about the pros ad cons of the use of this technique in the future. Should it be used and why or why not? What is the availability of this test? cost? What could it mean to the future of magnesium research, which is lacking an accurate, non-invasive test for magnesium status.
Response: To accommodate the comment of the reviewer we added the following text to the discussion section of the revised version of the manuscript “As a standalone test, the cost for NMR-assay for Mg would be comparable to a lipid panel derived from the NMR assay. If ordered together with NMR LipoProfile test or NMR Extended Lipid Panel test, it would only add a small incremental cost since no additional NMR time would be needed. Unlike the chemical assays for Mg, the non-invasive NMR assay doesn’t require reagents, and one can get results for all of the NMR based tests from one NMR spectra obtained from a single draw of a specimen, adding to the overall cost saving. It is therefore of particular interest for research when samples are precious and available sample volume relatively low, as for instance often is the case in cohort studies and intervention studies using material of underlying biobanks. The assay is currently available through the NMR Global Research Services group at LabCorp for research use only. Investigators who have used the NMR LipoProfile test over the years for various cardiovascular, diabetes, nutrition and diet, and inflammation related research studies and clinical trials have the opportunity of retrospectively analyzing the stored NMR data for plasma Mg levels. Though the data presented in this paper pertains to EDTA plasma specimens, the NMR-assay for Mg can easily be adapted to work on serum specimens by sufficiently modifying the EDTA concentration in the NMR diluent.”

Question 7
Inclusion/exclusion criteria may identify additional limitations based on the study population included (i.e. Dutch population only).

Response: We agree with the reviewer that this is a limitation of the study. We therefore added the words “and of Dutch origin” into Line 303.

Question 8

Conclusion-What are the authors recommendations in terms of use of this technique for identifying magnesium status in the future?

Response: The NMR-assay for ionized Mg has the potential to be used as a screening test for patients with hypomagnesemia and hypermagnesemia and to monitor magnesium levels during dietary and drug interventions. Accordingly, we added the following new text to the conclusions section starting at the end of Line 312:

“Measurements of circulating Mg are currently used by physicians to identify patients with hypomagnesemia and hypermagnesemia. Hypermagnesemia or high magnesium may be an indication of renal impairment or failure or in patients with Addison disease. Marked increases in circulating Mg may also be found in patients taking Mg salts, such as those found in antacids, or in pregnant women with preeclampsia who are taking Mg sulfate as an anticonvulsant. Hypomagnesemia can occur in subjects who are on long-term hyperalimentation, intravenous therapy, alcoholism and other types of malnutrition or malabsorption. Mg deficiency has been shown to be associated with cardiac arrhythmias. This study suggests that low plasma Mg may also be useful for assessing the risk of T2DM in women.”

Reviewer 2 Report

Please see below:

1) Lines 52-54: how about any meta-analysis of cohort or randomised controlled trials?

2) Line 82: Please provide the inclusion and exclusion criteria. Citing a reference and asking the readers to refer all the details elsewhere is inappropriate.

3)Line 138: why the number of cigarettes was used to categorise smoker, non-smoker and former smokers? Any citation for this?

4) Results: Why there were no dietary intake results for Mg?

5) the authors dicussed about dietary Mg in introduction, but there were no results of dietary Mg.

6) line 75: how much would the assesment cost ?

7) another limitation is that the study findings were only application in Dutch population?

8)Did the authors correct for other confounders such as CVD and other chronic diseases in their analysis?

9) please improve the English throughout the text.

10) Were the authors included any certified reference materials for Mg to check for the accuracy and precision of Mg measured?

Author Response

Question 1
Lines 52-54: how about any meta-analysis of cohort or randomised controlled trials?

Response: We want to thank the reviewer for this suggestion. To accommodate the comment of the reviewer, we added mentioning of a meta-analysis of 11 prospective cohort studies in the introduction section, which shows inverse associations between circulating magnesium and incidence of hypertension, CHD and T2DM (Line 53, reference 13). A meta-analysis of randomised controlled trials was already mentioned in Lines 52-54.

Question 2
Line 82: Please provide the inclusion and exclusion criteria. Citing a reference and asking the readers to refer all the details elsewhere is inappropriate.

Response: We agree with the reviewer that the inclusion and exclusion criteria could have been mentioned more extensively. Accordingly, we added the sentence: “Those who were unable or unwilling to participate, pregnant women, and individuals using insulin were not allowed to participate” to the methods section of the revised version of the manuscript (Line 85-87). In addition, we added a consert diagram of the PREVEND study to visualize the consequences of the inclusion and exclusion criteria for the present study (Figure 1).

Question 3
Line 138: why the number of cigarettes was used to categorise smoker, non-smoker and former smokers? Any citation for this?

Response: We thank the reviewer for noting this. We stand corrected for mentioning numbers of cigarettes. Smoking behavior was categorized into the following categories: never smoker, former smoker or current smoker. To accommodate the comment of the reviewer, we removed mentioning of numbers of cigarettes from the methods section of the revised version of the manuscript and added citation to previous manuscripts in which we used the same categorization (Line 151-152).

Question 4
Results: Why there were no dietary intake results for Mg?

Response: No food frequency questionnaires or other dietary questionnaires were taken at the time of data collection. We therefore have no data on dietary magnesium intake in the PREVEND study.  

Question 5
The authors dicussed about dietary Mg in introduction, but there were no results of dietary Mg.
Response: To accommodate the comment of the reviewer we removed the part from the introduction where dietary Mg was mentioned.

Question 6
Line 75: how much would the assesment cost ?
Response: As a standalone test, the cost for NMR-assay for Mg would be comparable to a lipid panel. If ordered together with NMR LipoProfile test or NMR Extended Lipid Panel test, it would only add a small incremental cost since no additional NMR time would be needed. To accommodate the comment of the reviewer (and comment #6 of reviewer #1), the following text was added to the discussion section of the revised version of the manuscript:
“As a standalone test, the cost for NMR-assay for Mg would be comparable to a lipid panel derived from the NMR assay. If ordered together with NMR LipoProfile test or NMR Extended Lipid Panel test, it would only add a small incremental cost since no additional NMR time would be needed. Unlike the chemical assays for Mg, the non-invasive NMR assay doesn’t require reagents, and one can get results for all of the NMR based tests from one NMR spectra obtained from a single draw of a specimen, adding to the overall cost saving. It is therefore of particular interest for research when samples are precious and available sample volume relatively low, as for instance often is the case in cohort studies and intervention studies using material of underlying biobanks. The assay is currently available through the NMR Global Research Services group at LabCorp for research use only. Investigators who have used the NMR LipoProfile test over the years for various cardiovascular, diabetes, nutrition and diet, and inflammation related research studies and clinical trials have the opportunity of retrospectively analyzing the stored NMR data for plasma Mg levels. Though the data presented in this paper pertains to EDTA plasma specimens, the NMR-assay for Mg can easily be adapted to work on serum specimens by sufficiently modifying the EDTA concentration in the NMR diluent.”

Question 7
Another limitation is that the study findings were only application in Dutch population?
Response: We agree with the reviewer that including only Dutch individuals is a limitation of this study. Therefore, we added the following to the Discussion section of the manuscript at the end of Line 303: “..and of Dutch origin”.

Question 8
Did the authors correct for other confounders such as CVD and other chronic diseases in their analysis?
Response: Covariates for the Cox Proportional-Hazards models were chosen based on literature and on univariate associations with NMR magnesium that we found in our Baseline table (Table 2). To accommodate the comment of the reviewer (and comment #5 of reviewer #1), we have performed sensitivity analyses in which we replaced adjustment for antihypertensive treatment in Cox regression analyses by adjustment for CVD and in which we replaced adjustment for eGFR in Cox regression analyses by adjustment for CKD (Table 4). These adjustments for the presence of CVD and CKD did not materially change our findings. To further accommodate the comment of the reviewer, we have added mentioning of performance of these sensitivity analyses to the methods section of the revised version of the manuscript and added mentioning of the results of these analyses to the results section of the revised version of the manuscript (Lines 184-187).

Question 9
Please improve the English throughout the text.
Response: Our native English speaking co-authors improved the English throughout the text.  

Question 10
Were the authors included any certified reference materials for Mg to check for the accuracy and precision of Mg measured?
Response: The Magnesium concentrations were standardized against a 25.0 mM solution of ACS Reagent Grade MgCl2.6H2O (MilliporeSigma, US). Defined amounts of the standard MgCl2 solution were spiked into dialyzed serum devoid of ionized Mg.  Accuracy was ascertained through recovery experiments done from 0 to 4.0 mM Mg concentrations. Similar to other NMR assays on Vantera, the commercial assay would also involve running 2 levels of serum controls serving daily check on accuracy and guarding against drift with time. The precision data on 2 serum pools is given on Table 1. To accommodate the comment of the reviewer, we added the following text to the  Methods section at end of Line 123:
“The Mg concentrations were standardized against a 25.0 mM solution of ACS Reagent Grade MgCl2.6H2O (MilliporeSigma, US). Defined amounts of the standard MgCl2 solution were spiked into dialyzed serum devoid of ionized Mg.  Accuracy was ascertained through recovery experiments done from 0 to 4.0 mM Mg concentrations. Similar to other NMR assays on Vantera, the commercial assay would also involve running 2 levels of serum controls serving daily check on accuracy and guarding against drift with time.”

 Reviewer 3 Report

The manuscript “Plasma Magnesium, Measured by Nuclear Magnetic Resonance Spectroscopy, and Risk of Developing Type 2 Diabetes Mellitus” by Schutten et al., provides an insight into the lower NMR-measured plasma ionized Mg was independently associated with a higher risk of developing T2DM in women, but not in men in a prospective Dutch cohort from PREVEND study. This manuscript is a well-written and very detailed experimental investigation of the given hypothesis.

However, although the findings are valuable, they are not well described or discussed. The manuscript needs minor modifications to improve the quality and visibility of the readers as follows.

The aim of the study is to develop and validate an assay for quantifying ionized Mg in plasma using NMR spectra with newly developed NMR-based assay in a large Dutch cohort study. It was comparable with Roche modular analyzer total Mg measurements. But in this study the interesting finding it was lower Mg is associated with a higher risk of developing T2DM in women, but not in Men. 

Hence it is good to include more discussion on Women Mg handling in physiology and in diabetes or pre/postmenopausal situations from available literature. 

Provide suitable title according to the outcome of the study; it is advisable to include gender difference and from which population (include Women and Dutch cohort phrase in the title)

Author Response

1) The manuscript “Plasma Magnesium, Measured by Nuclear Magnetic Resonance Spectroscopy, and Risk of Developing Type 2 Diabetes Mellitus” by Schutten et al., provides an insight into the lower NMR-measured plasma ionized Mg was independently associated with a higher risk of developing T2DM in women, but not in men in a prospective Dutch cohort from PREVEND study. This manuscript is a well-written and very detailed experimental investigation of the given hypothesis.
Response: We want to thank the reviewer for the kind words.

2) However, although the findings are valuable, they are not well described or discussed. The manuscript needs minor modifications to improve the quality and visibility of the readers as follows.

The aim of the study is to develop and validate an assay for quantifying ionized Mg in plasma using NMR spectra with newly developed NMR-based assay in a large Dutch cohort study. It was comparable with Roche modular analyzer total Mg measurements. But in this study the interesting finding it was lower Mg is associated with a higher risk of developing T2DM in women, but not in Men.

Hence it is good to include more discussion on Women Mg handling in physiology and in diabetes or pre/postmenopausal situations from available literature.

Response: We agree with the reviewer that the discussion of the manuscript could be extended in such a way. To accommodate the comments of the reviewer, we added the following text with citing references to the discussion section of the revised version of the manuscript:

 “It has indeed been established that estrogen significantly affects renal magnesium handling (Groenestege WM et al. J Am Soc Nephrol 2006; 17: 1035-1043, van der Wijst J et al. Magnesium Research 2009; 22: 127-132), likely explaining higher circulating Mg levels in pre-menopausal women than in post-menopausal women and cycling of Mg levels in pre-menopausal women (Grochans E et al. Magnesium Research 2011; 24: 209-214, Muneyyirci-Delale O et al. Fertil Steril 1998; 69: 958-962). Low circulating Mg levels and T2DM are known to be related, mainly through insulin resistance rather than through insulin secretion, but cause and effect relationships remain to be established (Gommers LMM et al. Diabetes 2016; 65:3-13). Consistent with this, it has been observed that low circulating Mg levels are associated with diabetes, insulin resistance and obesity in women, but not in men (Bertinato J et al. Food Nutr Res 2015; 59: 25974). It has also been shown that there is a sex difference in the relationship of urinary magnesium excretion to glycaemic control in patients with T1DM (Brown IR et al. Clin Chim Acta 1999; 283: 119-128). Albeit not in the field of diabetes, it has repeatedly been suggested that there is a sex-difference in the prospective association of Mg intake and low circulating Mg levels with cardiovascular mortality, with associations predominantly present in or limited to women (Xu T et al. Int J Cardiol 2013; 167: 3044-3047, Velat I et al. Int J Cardiol 2013; 168: 4437-4438, Ye H et al. 2018; 120: 415-423).       

3) Provide suitable title according to the outcome of the study; it is advisable to include gender difference and from which population (include Women and Dutch cohort phrase in the title).
Response: To accommodate the suggestion of the reviewer, we changed the title of the revised manuscript. The modified title is: “Plasma Magnesium, Measured by Nuclear Magnetic Resonance Spectroscopy, is Associated with Risk of Developing Type 2 Diabetes Mellitus in Women: Results from a Dutch Prospective Cohort Study”.